# On Event Individuation for Document-Level Information Extraction

**William Gantt**[1]    **Reno Kriz**[*2]    **Yunmo Chen**[*2]
**Siddharth Vashishtha**[*1]    **Aaron Steven White**[1]
[1] University of Rochester  [2] Johns Hopkins University
{wgantt@ur.|svashis3@ur.|aaron.white@}rochester.edu, {yunmo|rkriz1}@jhu.edu

## Abstract

As information extraction (IE) systems have grown more adept at processing whole documents, the classic task of *template filling* has seen renewed interest as a benchmark for document-level IE. In this position paper, we call into question the suitability of template filling for this purpose. We argue that the task demands definitive answers to thorny questions of *event individuation* — the problem of distinguishing distinct events — about which even human experts disagree. Through an annotation study and error analysis, we show that this raises concerns about the usefulness of template filling metrics, the quality of datasets for the task, and the ability of models to learn it. Finally, we consider possible solutions.

## 1 Introduction

*Template filling* involves extracting structured objects called *templates* from a document, each of which describes a complex event and its participants.[1] The task comprises both *template detection* — determining the number of templates present in a document and their types — as well as *slot filling* or *role-filler entity extraction* (REE) — assigning extracted entities to *slots* in those templates that characterize the roles entities play in the event. Since documents routinely describe multiple events, documents in template filling datasets regularly feature multiple templates, often of the same type. Correctly *individuating* events is thus a crucial and challenging component of the task. Here, we argue that whereas there is often more than one reasonable way to individuate events, template filling admits no such pluralism, which undermines its value as a benchmark for document-level IE. To support our claim, we draw evidence from several sources:

> *A bomb exploded today in a Lima restaurant, and a second device that had been placed in the same establishment was deactivated by the Peruvian National Police.* There were no victims, and the explosion caused very little damage to the restaurant…Guerrillas of the Tupac Amaru Revolutionary Movement (MRTA) have claimed credit for the terrorist act….

Figure 1: An excerpted document from MUC-4. Does this describe one bombing or two? We argue that this question is confusing and that disagreements about the "right" answer cause problems for document-level IE.

- A demonstration of how template filling *evaluation metrics* draw arbitrary distinctions between nearly identical system predictions that differ only in their individuation decisions.

- A (re-)annotation study, whose results suggest that template filling *datasets* reflect questionable individuation judgments about which even human experts may disagree — and even when given annotation guidelines.

- An error analysis on recent template filling *models*, showing that they struggle to learn the individuation rules latent in these datasets.[2]

Finally, we conclude by considering how the IE community might address these issues.

## 2 Background

**Template Filling**   Template filling is arguably *the* foundational task of IE (Grishman, 2019), with its origins in the Message Understanding Conferences (MUCs) of the 1990s (Sundheim, 1992; Grishman and Sundheim, 1996). These evaluations produced the MUC-4 dataset, which remains a primary benchmark for both template filling (Cham-

---

*Equal contribution.

[1]Following standard usage, we use "template" to denote an *instance* of a template *type* (cf. Chen et al. (2023)).

[2]All data and analysis code is available here: https://github.com/wgantt/event_individuation.

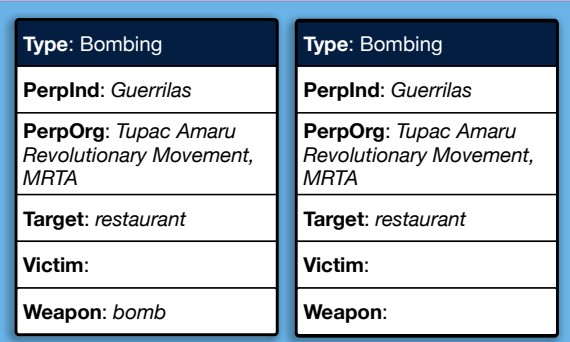

Figure 2: Gold MUC-4 templates for the document shown in Figure 1. Plausibly, the document describes just a single bombing (left template), yet predicting only this template would carry a substantial penalty in score (CEAF-REE F1 = 57.14%), even though it recovers all annotated entities and assigns them the correct roles.

bers and Jurafsky, 2011; Du et al., 2021b; Das et al., 2022; Chen et al., 2023) and for the REE subtask (Patwardhan and Riloff, 2009; Huang and Riloff, 2011, 2012; Huang et al., 2021; Du et al., 2021a). MUC-4 focuses on incidents of political violence in Latin America and presents an ontology of six template types, all with the same five slots.

More recently, the IARPA BETTER program has released the BETTER Granular dataset (Soboroff, 2023), which has begun to receive attention (Holzenberger et al., 2022; Chen et al., 2023). BETTER also features six template types, but unlike MUC-4, each template has a distinct set of slots, which may have other types of fillers beyond entities. We take both datasets as case studies.[3]

**Event Individuation** At the core of template filling is the question of *event individuation*: systems must determine when one event (i.e. template) is being described or several. This is a profoundly difficult problem that long antedates the field of IE, with a rich history in philosophy and linguistics (Lemmon, 1967; Kim, 1973; Quine, 1985; Cleland, 1991; Davidson, 2001, *i.a.*). At this level, the problem is well beyond IE's purview, as IE systems plainly should not be made to take a hard line on vexed metaphysical questions.

Instead, for IE purposes, we might hope to find some *rules of thumb* for event individuation that agree broadly with human judgments, and that can be consistently applied during annotation. Work on template filling itself has had relatively little to say on such rules, although we highlight some relevant discussion from the MUC-4 proceedings

in Appendix A. Rather, the fullest treatment of this subject comes from work on event coreference by Hovy et al. (2013), who posit conditions for full event *identity* — the inverse of the individuation problem. Appealing to the notion of a *discourse element* (DE), an event or entity that is referenced in the text, they stipulate that "two mentions fully corefer if their activity/event/state DE is identical in all respects, as far as one can tell from their occurrence in the text. (In particular, *their agents, location, and time are identical or compatible*)."

Differences in times, locations, and "agents" (i.e. participants) between two event descriptions are intuitively each sufficient conditions for individuating events and, importantly, accord with the guidelines for individuating templates provided for MUC-4 and for BETTER (see Appendix A). Template *types* offer another, coarser means of distinguishing events. Here, we focus analysis on individuation *within type*, since this is where most of the difficulty lies, and consider whether these guidelines are adequate to enable annotators and models to converge on similar individuation decisions.

## 3 Individuation and Evaluation

To show why individuation matters for template filling, we first consider how the task is evaluated. In contrast to events in event extraction, which are associated with lexical triggers, templates are not anchored to specific lexical items. Given a predicted template, there are often multiple gold templates one could use as a reference, and one must therefore establish an *alignment* ($A$) between them. All template filling metrics select the alignment that produces the highest total score, summed over aligned reference and predicted templates $T_R$ and $T_p$, given a template similarity function $\phi$:

$$A = \underset{A'}{\arg\max} \sum_{(T_R, T_P) \in A'} \phi(T_R, T_P) \qquad (1)$$

and given the constraint that each reference template is aligned to at most one predicted template with matching type.[4] Metrics differ mainly in their choice of $\phi$. It is this reliance on alignments that makes individuation so critical: *If individuation decisions in the reference templates are ill-grounded, this can yield punishingly large differences in score for otherwise semantically similar predictions.*

---

[3]Details on each dataset can be found in Appendix A.

[4]With this constraint, finding the optimal alignment is thus a maximum bipartite matching problem. See Appendix B.

We illustrate this with the example document from MUC-4 shown in Figure 1 and the gold templates in Figure 2.[5] Although arguably only one bombing is described in the text, the annotations feature two templates — one for the bomb that detonated and one for the bomb that was deactivated.[6] However, a system that predicts only the left template will be heavily penalized for having "missed" the one on the right, even though it identifies all relevant entities and assigns them the correct roles. Under the standard alignment-based metric for MUC-4, CEAF-REE (Du et al., 2021a), such a system would receive a score of only 57% F1.

Thus, because systems are tasked with reproducing *particular* (often idiosyncratic) individuation decisions, distinctions without much semantic difference can reverberate as significant differences in score. Alignment-based metrics therefore present a serious obstacle *in principle* to meaningful model comparison. But in practice, we will see that models struggle to individuate effectively *at all* (§5).

## 4 Individuation Replicability

One could argue that the metrics *per se* are not the problem, but rather questionable individuation decisions embodied in the annotations. If those decisions reflect real differences between events, then the difference between predicting one template and predicting two plausibly *should* correspond to a large difference in score.

Accordingly, we next investigate whether these decisions *are* well-founded in MUC-4 and BETTER. To do so, we conduct a re-annotation study of a random subset of documents from each dataset and evaluate agreement between these new annotations and the original ones. If the original annotations reflect sound individuation decisions, then we would expect them to be recoverable with high accuracy by annotators versed in their ontologies.

Three of the authors of this paper, each an experienced IE researcher, reannotated a subset of documents from the MUC-4 and BETTER training splits for template detection only, indicating how many templates of each type are present in each document. Documents were selected via stratified sampling, with the goal of obtaining 10 documents annotated for each template type. Within type, we further stratified by number of templates. This

| BETTER | | | MUC-4 | | |
|---|---|---|---|---|---|
| **Type** | $\alpha$ | $\alpha^+$ | **Type** | $\alpha$ | $\alpha^+$ |
| **Corruption** | 0.24 | 0.27 | **Arson** | 0.58 | 0.35 |
| **Disaster** | 0.21 | 0.21 | **Attack** | 0.44 | 0.42 |
| **Epidemic** | 0.36 | 0.28 | **Bombing** | 0.82 | 0.66 |
| **Migration** | 0.20 | 0.14 | **Kidnapping** | 0.81 | 0.73 |
| **Protest** | 0.13 | 0.07 | **Robbery**[*] | 1.00 | 1.00 |
| **Terrorism** | 0.30 | 0.23 | | | |
| **Aggregate** | 0.25 | 0.22 | **Aggregate** | 0.63 | 0.56 |

Table 1: Krippendorff's $\alpha$ for IAA on the number of templates of each type for the reannotated MUC-4 and BETTER documents. $\alpha = 1$ indicates full agreement; $\alpha = 0$ is agreement at chance. [*]The sampling procedure excluded samples of forced work stoppage templates and yielded only three robbery templates, so robbery results should be taken with caution.

procedure yielded 42 documents for MUC-4 and 60 for BETTER.[7] Annotators were provided with complete descriptions of both ontologies, as well as relevant excerpts from the annotation guidelines. Importantly, these excerpts contained instructions for template individuation consistent with the criteria from Hovy et al. (2013) discussed in §2.

We consider the level of agreement among the three annotators alone ($\alpha$), and when additionally incorporating the gold annotations ($\alpha^+$). Table 1 presents nominal Krippendorff's $\alpha$ (Krippendorff, 2018) in aggregate and by template type. While agreement on some MUC-4 template types is reasonably strong among annotators alone, we should expect this across the board. Instead, we find that *aggregate* agreement is at best moderate for MUC-4 ($\alpha = 0.63$) and consistently low for BETTER ($\alpha = 0.25$). This is disheartening considering the annotators' qualifications and their access to the annotation guidelines. Most concerning, though, is that agreement is *even lower* when including the gold annotations ($\alpha^+ = 0.56, 0.22$). Thus, if human experts cannot reliably agree either among themselves or with the gold annotations, this suggests that the individuation problem is ill-posed.

## 5 Do Models Learn to Individuate?

Given these observations, we now ask what the consequences are for our IE models. Even if experts struggle to replicate the original individuation decisions, it is possible that the datasets exhibit some *internal* consistency, and that a model could therefore learn their template count distributions.

To explore this possibility, we consider three

---

[5]Other challenging examples are shown in Appendix C.

[6]We note the oddity of the missing annotation of *second device* as a **weapon**, but this is not important to the argument.

[7]Additional study details are in Appendix D.

recent top-performing models for template filling: ITERX (Chen et al., 2023), the current state-of-the-art; GTT (Du et al., 2021b), the previous best model; and TEMPGEN (Huang et al., 2021), a third competitive system. We investigate the predictions of all three systems on the MUC-4 test split, as well as the predictions of ITERX on the BETTER test split.[8] While some prior work has attempted a complete taxonomy of template filling model errors (Das et al., 2022), our concern here is exclusively with detection problems.

First, we note that all three models make a significant number of detection errors. Even the strongest detection model (GTT) still incorrectly predicts the total number of templates on 32% of MUC-4 test documents (Table 2, top row).

Second, we observe that the errors of all three models are overwhelmingly system *misses*: among documents with detection errors, ITERX, GTT, and TEMPGEN predicted fewer templates than the reference on 72.5%, 81.3%, and 80.0% of them, respectively (Table 2, second row). These findings are consistent not only with prior error analyses (Das et al., 2022),[9] but also with our observations thus far: if individuations are close to arbitrary from a model's perspective, it can generally do no better than to collate all entities relevant to a particular event type into a single template.

If this is in fact what these models are learning to do, then *collapsing* predicted templates into a single aggregate template per type (taking per-slot unions over their fillers) should not substantially alter overall scores. Doing this for all model predictions and recomputing scores bears this out: across the three models, there is an average drop of just over a single point in overall score on MUC-4 and a mere 0.3 points on BETTER (Table 2, bottom two rows). By comparison, collapsing all *gold* templates by type (not shown) yields a drop of 12.7 points in overall score ($100 \rightarrow 87.3$) on MUC-4 and 24.2 points on BETTER ($100 \rightarrow 75.8$).

# 6 Solutions

The value of template filling as an IE benchmark is undermined both by datasets that make dubious individuation decisions and by metrics that sharply penalize deviations from those decisions. So what

---

[8]BETTER predictions are available only for ITERX. Additional model details in Appendix E.

[9]See §7.2 of Das et al. 2022: "The main source of error for both early and modern models is missing recall due to missing templates and missing role fillers."

---

|  | BETTER | MUC-4 | | |
|---|---|---|---|---|
|  | ITERX | ITERX | GTT | TEMPGEN |
| \|gold\| $\neq$ \|pred\| (%) | 34.4 | 34.5 | 32.0 | 37.5 |
|   of which \|gold\| > \|pred\| (%) | 72.7 | 72.5 | 81.3 | 80.0 |
| Temp. Precision | 89.7 | 79.5 | 81.1 | 82.7 |
| Temp. Recall | 74.5 | 59.7 | 57.7 | 57.2 |
| Temp. F1 | 81.4 | 68.2 | 67.4 | 67.6 |
| Combined Score (original) | 30.0 | 53.0 | 50.2 | 47.2 |
| Combined Score (collapsed) | 29.7 | 51.4 | 48.8 | 46.9 |

Table 2: Test set metrics for template filling models. Combined scores use CEAF-REE for MUC-4 and the official Granular score for BETTER. "|gold|" and "|pred|" denote number of gold and predicted templates.

can be done? Here, we briefly consider several possible solutions, each with different tradeoffs.

## 6.1 Evaluation Aggregated By Type

One possibility is simply to make standard the evaluation setting from §5 in which templates are aggregated by type. This presents a clearer picture of whether systems are getting the big things right (distinguishing events of different types), but it gives up on difficult within-type individuation.

## 6.2 Multiple Reference Individuations

A second possibility is to annotate all plausible individuations, evaluating a system's predictions against the reference individuation that maximizes the score of those predictions — just as template filling metrics use the score-maximizing template alignment (§3). This has the advantage of accommodating the natural plurality of ways events may be individuated without privileging any one of them. But it has the obvious *dis*advantage of requiring much more annotation. Moreover, although it may often be clear what the set of "plausible" individuations is, this is by no means always teh case. For one, any event ontology will present irreducibly hard questions about whether particular event descriptions meet the criteria for being labeled with a particular type. For another, *distributivity* — whether and how an event distributes across its participants — can present additional challenges. For instance, from (1), we can infer that Alice, Bob, Cam, and Dana are *each* abducted. But do we really want to say that all 15 possible ways of grouping them into kidnapping templates are valid? Further, how should we handle cases involving *aggregate participants*, which may imply multiple events, but where the text offers no basis for distinguishing them, as in (2)?

(1)    Alice, Bob, Cam, and Dana were abducted.

(2)    Four residents were abducted in two days.

Such considerations may make aggregate evaluation (§6.1) look more attractive. But using multiple references is not an all-or-nothing solution: some plurality in annotation is certainly better than none.

## 6.3 Subevent-Based Evaluation

A shortcoming of both §6.1 and §6.2 is that they largely side-step the issue of *why* a given individuation might be appealing. Particular individuation decisions are often motivated by important and uncontroversial facts about events, and whether an annotator makes a given decision hinges on how salient they believe those facts to be. To return to Figure 1, there is no dispute that a second bomb was deactivated by police. Rather, the dispute is about whether this is salient in a way that merits annotating a second bombing event. A perfectly reasonable reply to this question is: *who cares?* Presumably, what we actually care about as readers is *that* a second bomb was deactivated — not how best to accommodate this fact in our ontology.

These considerations motivate a third solution: to focus instead on extraction of *fine-grained* events — *subevents* of the coarse-grained events captured in ontologies like MUC-4 and BETTER. The advantage of this approach is that it eliminates a major source of disagreement about individuation — namely, the often unclear conditions that must be met for coarse-grained events to obtain. A bombing-related subevent ontology, for instance, might have a `detonation` subevent, whose conditions for obtaining are clearer by virtue of being more specific. We imagine that such ontologies would also cover the full gamut of activities that may occur before (e.g. device placement) and after (e.g. perpetrators' escape) the primary subevent(s) (e.g. detonation). In this way, subevent ontologies resemble *scripts* (Schank and Abelson, 1977) or *narrative schemas* (Chambers and Jurafsky, 2008, 2009), and can capture much richer information about the temporal structure of events.

Moreover, focusing on subevents may render document-level event extraction more amenable to methods from a now well-established line of work that recasts (typically sentence-level) event extraction as question answering (QA; Liu et al., 2020; Du and Cardie, 2020; Li et al., 2020; Holzenberger et al., 2022) — methods that have become increasingly popular as the reading comprehension abilities of large, pretrained language models (LLMs) have drastically improved. However, approaches in this vein appear to be underdeveloped from the perspective of document-level event extraction. Specifically, questions about events tend to be posed independently from each other and also tend not to query for some of the properties we have highlighted as being important to individuation, such as irrealis and distributivity.

Combining these threads, we envision an improved approach to QA for (document-level) event extraction that (1) is based on subevent ontologies, (2) poses questions about individuation-relevant properties, and (3) is conditional, in the sense that (a) questions about later subevents are conditioned on questions (and answers) about earlier ones and that (b) questions about roles are conditioned on a relevant subevent being identified. This would allow IE researchers and end users to leverage the capabilities of LLMs to extract rich, natural language descriptions of events while dispensing with some of the individuation problems that we have seen arise with traditional template filling. Crucially, this could be evaluated *as QA*, without the need for template alignments (though questions must be associated with particular subevents). We illustrate this proposal in Appendix F with an example subevent ontology for `bombing` events and actual outputs from a QA dialogue with ChatGPT.[10]

We stress that our proposal would not solve all problems raised here: most of the difficulties we highlight at the event level could *in principle* arise at the subevent level. Rather, our expectation is that they will do so much less often *in practice* and that subevents can enable much more detailed information about the broader events they constitute.

## 7 Conclusion

This work has considered the problem of *event individuation* — how to distinguish distinct events — and its ramifications for document-level event extraction. We have shown both that models for this task struggle to replicate gold event counts (§5) and, more fundamentally, that even human experts frequently make different individuation decisions (§4). These considerations motivate alternative modes of evaluation for this task (§6), which we hope are adopted in future work. More generally, in the spirit of Hovy et al. (2013), we hope that our work reminds NLP researchers of the linguistic and conceptual complexity of events, and of the need to take this complexity seriously.

---

[10] https://openai.com/blog/chatgpt

## Limitations

Some researchers have framed structurally similar IE tasks — notably, $N$-ary relation extraction — as template filling (Huang et al., 2021; Chen et al., 2023; Das et al., 2022). However, since this paper concerns classic template filling, which focuses on *events* (Jurafsky and Martin, 2021), the issues we raise about event individuation do not obviously extend to relations. Individuation of relations has its own challenges, but insofar as relations are more abstract than events and not spatially or temporally localized, they may be more readily individuable solely on the basis of the entities they involve.

Further, our analysis in general does not apply to other document-level IE tasks — notably, (multi-sentence) *argument linking* (also called *event argument extraction* or *implicit semantic role labeling*; O'Gorman, 2019; Ebner et al., 2020; Gantt, 2021; Li et al., 2021, i.a.) — where events are, per the task definition, distinguished on the basis of lexical triggers. In these cases, the questions we raise about individuation criteria do not arise.

## Ethics Statement

We do not believe this work raises any ethical issues beyond those already embodied in the data and models we analyze, all of which were introduced in earlier work. However, that is not to dismiss those issues. Beyond standard disclaimers about undesirable biases of LLMs (e.g. Brown et al., 2020), we note that MUC-4's focus on incidents of violence in Latin American nations (namely, Argentina, Bolivia, Chile, Colombia, Ecuador, El Salvador, Guatemala, Honduras, and Peru) may be liable to engender model biases that reflect negative stereotypes about people from or living in those countries. The BETTER corpus is significantly more diverse in its subject matter and appears less likely to bias models in a similar way, though we do not have empirical results to support this claim.

## Acknowledgments

We would like to thank Tim McKinnon, Marc Vilain, and members of the Center for Language and Speech Processing at Johns Hopkins for helpful questions and comments on this work. This work was supported by IARPA BETTER (2019-19051600005) and NSF-BCS (2040831). The views and conclusions contained in this work are those of the authors and should not be interpreted as necessarily representing the official policies, either expressed or implied, or endorsements of IARPA or the U.S. Government. The U.S. Government is authorized to reproduce and distribute reprints for governmental purposes notwithstanding any copyright annotation therein.

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

## A  Dataset Details

This appendix provides more detailed information on the MUC-4 and BETTER Granular datasets and on their guidelines for individuating templates.

### A.1  MUC-4

#### A.1.1  Ontology and Dataset Statistics

|                       | Train  | Dev | Test |
|-----------------------|--------|-----|------|
| documents             | 1,300  | 200 | 200  |
| documents w/ templates| 700    | 116 | 126  |
| total templates       | 1,114  | 191 | 209  |
| attack                | 636    | 97  | 127  |
| bombing               | 298    | 50  | 55   |
| kidnapping            | 114    | 31  | 14   |
| arson                 | 43     | 8   | 3    |
| robbery               | 17     | 3   | 1    |
| forced work stoppage  | 6      | 1   | 4    |

Table 3: Summary statistics of the MUC-4 dataset. Included in the "total templates" figures are several templates that are cross-classified as both `attack` and `bombing`. There is one such template in dev and five in test.

MUC-4 features six template types: `Arson`, `Attack`, `Bombing`, `Forced Work Stoppage`, `Kidnapping`, and `Robbery`. All template types share the same set of five slots: `PerpInd` (an individual perpetrator), `PerpOrg` (a group or organizational perpetrator), `Victim` (sentient victims of the incident), `Target` (physical objects targeted by the incident), and `Weapon` (weapons employed by the perpetrators).[11] All five slots accept entity-valued fillers, represented by sets of coreferent entity mentions, though systems need only predict one of those mentions to receive full credit. The type of each template instance is specified as a sixth slot, `incident type`. Complete dataset statistics are provided in Table 3. We use the preprocessed version of MUC-4 available here: `https://github.com/xinyadu/gtt/tree/master/data/muc`. The original (raw) data is available here: `https://www-nlpir.nist.gov/related_projects/muc/muc_data/muc_data_index.html`.

#### A.1.2  Multi-Template Annotation

Below is an excerpt from the MUC-4 dataset documentation, describing the conditions under which multiple templates are to be annotated for a document. This excerpt, along with those included in the instructions for the reannotation study (see Appendix D), were drawn from nn- (1992). Note that the criteria for annotating multiple instances here closely align with those presented in §2 from Hovy et al. (2013). Note also that although the `location`, `date`, and `category` slots are annotated in the original data, they are not included in the evaluated templates. The `location` and `date` slots are self-explanatory; the `category` slot indicates whether or not the event is an instance of "state-sponsored violence," meaning that the perpetrator (`PerpInd` / `PerpOrg`) is (a member of) the government, military, or police.

> If an article discusses more than one instance of the same relevant type of terrorist incident, each such incident should be captured in a separate template. **A "separate instance" is one which has a different filler for the location, date, category, or perpetrator slot.** If no distinctions of these kinds are made in the text, the instances should be incorporated into a single template, e.g., if the article describes "attacks" by "rebels" on "ten power stations" and does not give varying information on the location of those targets, etc. Note that the level of granularity is defined by

---

[11]These slots are the ones researchers have focused on since the end of the original MUC-4 evaluation, though the full ontology actually features 24 slots, of which these five (as well as the `incident type` slot) are a subset. See either of the data links above for more details.

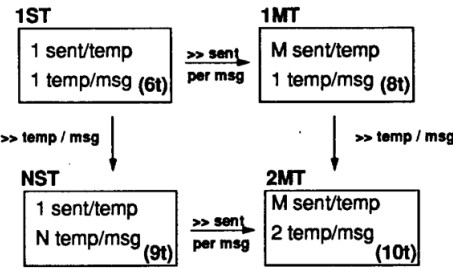

Figure 3: Four subsets of the MUC-4 test data evaluated in Hirschman (1992), reproduced exactly from that work.

> what can go in a slot; thus, an article that describes two bombings on targets on different streets in the same neighborhood should be captured in one template rather than two, since the location slot cannot have a filler that is at the level of granularity of a street.

The bombing example discussed in this passage is particularly interesting when compared to the bombing described in Figure 1 and annotated in Figure 2. According to the principle outlined here, it certainly seems as though the document in Figure 1 should have a single `bombing` template annotated, rather than two, as no distinctions between the annotated bombings can be drawn on the basis of the `location`, `date`, `category`, `PerpInd`, or `PerpOrg` slots.

Alongside `location`, `date`, and `category`, an irrealis-focused `stage of execution` slot was annotated for the original data, and indicated whether an event was `accomplished`, `attempted`, or only `threatened`. This is the basis for the distinction between the templates in Figure 2. The existence of these annotations is good news in that they may help in a recasting effort of the sort proposed in §7, but bad news in that they may form the basis for an individuation (e.g. Figure 2) without actually being part of the task.

### A.1.3 Event Individuation in the MUC-4 Proceedings

Beyond the quoted passage above, there is not much substantive discussion of individuation criteria in the MUC-4 proceedings themselves (muc, 1992) — and certainly not at the level of analysis we aim to provide in this work. However, both participants and organizers recognized the need for a principled *technical* approach to individuating events. Some of this discussion focuses on the closely related difficulty of defining what counts as a *relevant* terrorist incident. For instance, in considering shortcomings of the task, Sundheim (1992) notes: "The definition of a 'relevant terrorist incident' was inadequate in several respects. The distinction between a terrorist incident and other kinds of violent events — guerrilla actions, shootings of drug traffickers, etc. — is a difficult one to express in comprehensive, hard and fast rules."

Much of the proceedings discussion on this subject focuses purely at a practical level, describing the individuation techniques and heuristics different teams applied — typically under the heading of "discourse analysis" or "discourse processing." For instance, the NYU team's PROTEUS system adopted a strategy used by many teams, which involved detecting templates at the sentence level and then attempting to merge them according to certain criteria (Grishman et al., 1992). For NYU, these criteria included (among others) merging (1) when events affected the same target; (2) when events were described in the same sentence; or (3) when one event described an attack and the next event described an effect.

Another example of this practical focus is Hirschman (1992), who presents a study of systems' performance on different subsets of the test data, created based on (1) the number of templates in a document, and (2) the number of sentences required to fully populate a given template. The cross product of these variables defines the four subsets of the test data shown in Figure 3. The aim of the study was to evaluate so-called "lazy merger" problems (i.e. *overgeneration*, or failing to merge templates that *ought* to be merged) and "greedy merger problems" (i.e. *undergeneration*, merging templates that *ought not* to be merged). The four subsets were defined to target examination of particular combinations of these two problems, with the "1MT" documents being subject only to lazy merger; "NST" documents being subject only to greedy merger; and "1ST" and "2MT" sets being subject to both. Interestingly, she observes that on average, overall score was worst for systems on the 1ST subset (with only one template

|                      | Phase I |     |      | Phase II |     |      |
|----------------------|---------|-----|------|----------|-----|------|
|                      | Train   | Dev | Test | Train    | Dev | Test |
| documents            | 94      | 20  | 20   | 168      | 34  | 32   |
| documents w/ templates | 88    | 19  | 19   | 168      | 34  | 32   |
| templates            | 252     | 56  | 47   | 255      | 57  | 57   |
| corruption           | 63      | 10  | 11   | 3        | 0   | 0    |
| disaster[*]          | 0       | 0   | 0    | 68       | 11  | 11   |
| epidemic             | 31      | 7   | 1    | 76       | 23  | 23   |
| migration[*]         | 0       | 0   | 0    | 81       | 20  | 20   |
| protest              | 109     | 37  | 25   | 12       | 2   | 2    |
| terrorism            | 49      | 2   | 10   | 15       | 1   | 1    |

Table 4: Summary statistics of the Phase I and Phase II BETTER Granular datasets. [*] Denotes that this template type is part of the Phase II ontology only.

per document), suggesting that relevancy judgments (at least at the time) were an even greater challenge than individuation. However, the subsets are quite small and the results should be interpreted with caution.

### A.2 BETTER

#### A.2.1 Ontology and Dataset Statistics

We use the BETTER Granular Phase I and Phase II datasets, which collectively feature six template types that exhibit a greater diversity of topics than the MUC-4 templates: `Corruplate` (incidents of corruption), `Disasterplate` (natural disasters), `Displacementplate` (human migration events), `Epidemiplate` (epidemics), `Protestplate` (protests), and `Terrorplate` (acts of terrorism). While some slots are common to two or more of these template types, most slots are unique to each type, reflecting event-specific roles, similar to those in FrameNet (Baker et al., 1998). Complete dataset statistics are provided in Table 4. We use the versions of the data provided here: https://ir.nist.gov/better/.

#### A.2.2 Multi-Template Annotation

As the BETTER documentation has not been made public, we are unable to quote from it, though we will attempt to summarize the relevant considerations for multi-template annotation. Like in MUC-4, the BETTER documentation indicates that multiple templates may be annotated when the times or locations of the events described differ from each other. Some irrealis and time annotations are provided, but these are associated with specific fillers of event-valued slots, and are not used to distinguish whole templates, as the `stage of execution` slot is for MUC-4. Multiple templates of the same type may also be required when those templates apply to different people (e.g. different individuals accused of corruption). However, this criterion is somewhat opaque, as many templates feature multiple individuals as fillers of the same slot, and it is unclear exactly when this condition merits individuating a given set of events.

## B Evaluation Metrics

Due to space limitations, definitions of the alignment-based evaluation metrics for MUC-4 and BETTER are omitted in the main text. We refer the reader to Appendix D in Chen et al. (2023) for thorough descriptions and discussion of these metrics, but will briefly present them here.

In §3, we observe that all template filling metrics determine an alignment between system-predicted and reference templates that maximizes an overall score (Eq. 1), which decomposes as a sum over similarity scores for aligned template pairs $(T_P, T_R)$. Recall that this alignment is subject to two constraints: (1) At most one predicted template may be aligned to each reference template, and (2) aligned templates must be of the same type. Constraint (1) makes this a maximum bipartite matching problem, typically solved with the Kuhn-Munkres algorithm (Kuhn, 1955; Munkres, 1957). Note that $T_P$ can be null (for missed reference templates), as can $T_R$ (for spuriously predicted templates). In either case, $\phi(T_P, T_R) = 0$.

For the original MUC-4 evaluation, $\phi$ was essentially an F1 score over all slot fillers in a given template. A predicted slot filler counted as correct iff it was featured among the reference fillers for the corresponding slot in an aligned template.[12] All fillers of spuriously predicted templates were counted as false positives and all fillers of missed reference templates were counted as false negatives (Chinchor, 1992). As noted in footnote 11, the set of slots included not only those shown in Figure 2, but a number of others that were annotated but for which results have not been reported since the original evaluation.

More recent reporting on MUC-4 has used the CEAF-REE score (Du et al., 2021a,b; Huang et al., 2021; Chen et al., 2023). Introduced by Du et al. (2021a), CEAF-REE is similar to the slot-filler F1 score used in the original MUC-4 evaluation, except that evaluation is based only on the five slots in Figure 2, plus the template type (treated as a sixth slot). Chen et al. (2023) present a variant of CEAF-REE, dubbed CEAF-R*M*E, which corrects certain problems they identify with the original metric. To our knowledge, they are the only ones to have reported CEAF-RME scores.

The official scoring metric for BETTER differs from both the original MUC-4 slot filler F1 and CEAF-REE in that it reflects the *product* of slot-level and template-level F1 scores, conditional on the optimal alignment. This score does *not* decompose over template pairs and so cannot be directly optimized via maximum bipartite matching. Instead, the BETTER scorer optimizes a quantity called *response gain*: the difference between the number of correct and incorrect slot fills under a given alignment. Optimizing alignments for this objective is theoretically guaranteed to yield the optimal alignment under the official score within a small probabilistic error bound. For more details on this metric, see Chen et al. (2023).

## C   Additional Difficult Examples

Lest the example presented in Figure 1 appear cherry-picked, we present here additional examples of documents from both datasets that present challenging individuation decisions and that similarly strike us as lacking a single, obviously correct answer. Such examples are abundant.

### C.1   MUC-4 Examples

We note that the MUC-4 source texts is uncased; we have cased the ones below simply for readability.

**Example 1**
**Document Text:**

> Two vehicles were destroyed and an unidentified office of the Agriculture and Livestock Ministry was heavily damaged following the explosion of two bombs yesterday afternoon. The National Fire Department reported that following the explosions at 1830, a fire erupted that partially destroyed an unidentified office in the Agriculture and Livestock Ministry building. It was reported that two vehicles in the area where the bombs were detonated were destroyed and several houses in the area were damaged. The damaged offices are located at 123 A Avenue in the San José neighborhood in the western section of San Salvador. Fortunately, there were no casualties reported as a result of this incident, for which the FMLN guerrillas are being held responsible.

**Annotated Templates:** 1 bombing template
**Discussion:** This example provides an interesting foil for the one in Figure 1 and Figure 2. Here, just as in that earlier example, two bombs are involved. But unlike the earlier example, only a single template is annotated. From the text, it appears that the bombs were likely planted near each other, but so too were the pair of bombs in Figure 1 — in the same restaurant, in fact. The only distinguishing feature between these examples is the fact that in Figure 1, only one of the bombs actually detonated. This is an important fact, but whether one actual detonation and one non-detonation makes for two "bombings" is dubious and is hardly the only reasonable reading, as we discuss in the main text. Moreover, were the second bomb

---

[12] MUC-4 documentation draws a distinction between "string-fill" slots — those that are filled by entities, represented by a single canonical mention — and "set-fill' slots — those that are filled by one of a fixed set of possible values (Chinchor, 1992). Determining a match between system and reference slot-fillers for set-fill slots is straightforward. However, to determine a match for string-fill slots, the original evaluation applied various fuzzy string matching heuristics.

to actually have detonated, the annotation guidelines ([Appendix A](#)) imply that we would instead (again) have a single bombing, owing to the equivalence between all the relevant slots in the two templates. This is idiosyncratic at the very least.

**Example 2**
**Document Text:**

> Military sources reported today that fighting broke out on the night of 5 March in the war-torn Chalatenango department to the north of San Salvador, where the guerrillas suffered eight casualties. The Armed Forces Press Committee reported that the clashes took place near San Isidro Labrador, where three guerrillas were killed and five others were wounded. The Farabundo Marti National Liberation front (FMLN), which has been fighting the U.S.-backed army for the last 10 years, has traditionally maintained a high profile in strategic areas of Chalatenango. Meanwhile, urban guerrilla commandos harassed troops belonging to the 1st Infantry Brigade last night at the Mariona Penitentiary, located on the northern outskirts of San Salvador, apparently leaving no casualties. Atlácatl battalion troops, which maintain a counterinsurgency operation at the foothills of Guazapa Hill, to the north of San Salvador, confiscated war materiél that the guerrillas had hidden underground. In another turn of events, FMLN sappers who continuously sabotage the electrical power system destroyed several high-voltage posts in La Libertad and Cabanas departments in the central part of the country, and in Usulután department in the southeastern part of the country. There were widespread power outages in the three departments while technicians of the state-run electric power companies repaired the damage. The army and security corps deployed heavy patrols in San Salvador and in cities in the country's interior to try and counter the wave of attacks.

**Annotated Templates:** 3 `attack` templates
**Discussion**: For MUC-4, `attack` is a catch-all type to handle terrorist acts that are not adequately described by any of the other types, which makes it particularly challenging to identify. Furthermore, the documentation specifies that violent acts directed *against terrorists* are not to be annotated. This rule evidently excludes the eight guerrilla casualties described here, though it's not certain from the text that the guerrillas are terrorists. The three `attack` templates refer instead to the FMLN attacks on "high-voltage posts," with one template per location (department) in which posts were destroyed (La Libertad, Cabanas, and Usulután). This is sensible, though one of the annotators in our study (not unreasonably) considered these to be part of the same *coordinated* attack on the power system and annotated a single template.

### C.2 BETTER Granular Examples

**Example 1**
**Document Text:**

> President Cyril Ramaphosa has urged South Africans to "act responsibly" as the second wave of the Covid-19 pandemic gripped two provinces, with fears that it will sweep through the rest of the country during end of year celebrations.

> His worried Health Minister, Dr Zweli Mkhize, Saturday night 'raided' popular Cape Town bars and nightspots where merrymakers were breaking lockdown rules on social distancing, mask wearing and staying safe.

> What is bothering the South African government is the rapid rise in new cases in both the Eastern Cape Province - where the second wave of the virus has a full grip and is overwhelming - with as a similar pattern emerging in the Garden Route of the Western Cape, which borders the Eastern Cape, as well as in greater Cape Town, capital of the Western Cape.

> The fear is that the wave will prove unstoppable amid end of year celebrations and family gatherings which are bound to drive infections up further, threatening to overwhelm South Africa's already struggling medical system.

Overnight, South Africa recorded another 4,932 new Covid-19 infections and at least 160 more deaths. The pandemic has so far claimed over 21,000 lives.

Ramaphosa, speaking to the nation prior to his health minister's on-the-ground inspection of potential 'superspreader' events and venues, had announced a return to tougher emergency restrictions in the major urban area of the Eastern Cape, the Nelson Mandela Bay metro, where conditions are severe and from where nearly half of all new infections are coming.

But South Africa health authorities are moving away from the idea of sweeping new lockdowns, focusing instead on hotspots and attempting to stop the spread by restricting alcohol sales, imposing a tighter curfew between 10pm and 4am and reducing the numbers of people attending open-air and indoor venues.

**Annotated Templates:** 1 `Epidemiplate`
**Discussion:** This document describes just a single disease (Covid-19), but it's far from clear that it describes a single *epidemic*. In contrast to *pandemics*, which are global disease outbreaks, epidemics are geographically localized. Considering the individuation criteria for BETTER (Appendix A), we are justified in annotating multiple `epidemiplates` if there are disease outbreaks in distinct locations. The article describes outbreaks in both the Eastern Cape and Western Cape Provinces of South Africa. But is this one location (South Africa) or two (the two provinces)? Both seem reasonable. One could even argue that the emergence of the virus's second wave in the Garden Route and in Cape Town — both within the Western Cape Province – constitute separate epidemics of their own.

**Example 2**

**Document Text:**

Uganda's Foreign Affairs Minister Sam Kutesa and two other officials have denied corruption charges. Mr Kutesa, chief whip John Nasasira and junior labour minister Mwesigwa Rukutana appeared in court a day after they resigned. They are accused of abuse of office as well as financial loss over the 2007 Commonwealth summit in Uganda, in which scams allegedly cost some $150m (£95m). Former Vice-President Gilbert Bukenya was charged in July. He denies that he benefited from a $3.9m deal to supply cars used to transport dozens of heads of state during the summit. Mr Kutesa was also accused on Monday of taking large bribes from UK-based Tullow Oil. The minister and the company strongly denied the allegation. In court, prosecutor Sydney Asubo said the three had cost the government 14bn shillings ($4.8m; £3.1m). "The three irregularly convened a consultative cabinet meeting and decided that the government would fully fund the construction cost of driveways, parking areas and marina at Munyonyo Speke Resort," he said. They face up to 13 years in jail if convicted, reports the AFP news agency. Some MPs from Mr Bukenya's Buganda ethnic group had accused the government of selective justice by failing to prosecute anyone else. Last week, the Inspectorate of Government - the body charged with fighting corruption - said Mr Kutesa, Mr Nasasira and Mr Rukutana would be charged. President Yoweri Museveni said on Wednesday that the three officials had chosen to resign. "That's their decision because what we want is the truth," he told a news conference. Mr Museveni sacked Mr Bukenya in May as part of a cabinet reshuffle.

**Annotated Templates:** 4 `Corruplates`
**Discussion:** The article describes four individuals accused of corruption: Sam Kutesa, John Nasasira, Mwesigwa Rukutana, and Gilbert Bukenya. For this reason, it's understandable that the annotations would feature four corruption templates — one per person charged. However, the first three of the four seem to be implicated in the same corruption charges. Those charges could reasonably be counted as a single incident of corruption involving multiple actors, yielding two templates overall (or three if including Mr. Kutesa's bribery charge as a separate incident). This was a case in which there was *complete* disagreement among annotators in our reannotation study, who annotated two, three, and four `Corruplates`.

# D  Reannotation Study: Further Details

The reannotation study was designed and run by the first author, who did not participate in the annotation. The annotators were the three middle authors and are all fluent English speakers with strong IE backgrounds (§4). They worked independently over the course of several days and were not permitted to discuss any aspect of the task with each other or with the first author. Moreover, the authors did not discuss theories of, or approaches to, event individuation prior to beginning the study. Annotators were instructed to rely only on the guidelines provided in the study instructions. All annotation was unpaid.

In order to focus the task on within-type individuation, and to make it as straightforward possible, documents were required to have annotated gold templates of only a single type. As discussed in §4, we aimed to obtain 10 documents for each type, for each dataset. Given the single type constraint, this was not always possible — as was the case for the `robbery` and `forced work stoppage` templates in MUC-4 (see Table 1). For these cases, we selected as many documents as we could that satisfied the single type constraint, which resulted in three documents for `robbery`, and unfortunately none for `forced work stoppage`. Although it would be nice to have more documents for these types, we do not think this detracts from the general result that even experts struggle to agree on within-type individuation decisions.

The complete instructions for the MUC-4 reannotation study can be found in the GitHub repository associated with this paper (see footnote 2).

For BETTER, we selected documents for `Disasterplates` and `Displacementplates` from the Phase II data, as the Phase I data does not contain templates of this type. However, we selected all documents for `Corruplates`, `Epidemiplates`, `Protestplates`, and `Terrorplates` from the Phase I data, as the Phase II data had comparatively few documents annotated with these types. As the reannotation study instructions quote from the BETTER program documentation (which is nonpublic at the time of writing), we are not able to include these instructions.

Finally, when we say in §4 that we stratify by the number of templates within type, we mean that we first group all documents annotated exclusively with templates of type $T$ by the number of templates annotated, then sample round robin from each group, and uniformly at random within group.

In the main text, the (dis)agreement analysis uses the nominal version of Krippendorff's $\alpha$, which is equivalent to Fleiss's $\kappa$ (Fleiss, 1971) in settings like ours in which the same set of annotators annotate all items. This is meant to show the level agreement when considering only *whether* annotators agree on the number of events described, which is our main concern. By computing the *ratio* version of the $\alpha$ instead of the nominal, we may also take into account the *magnitude* of the difference in the number of annotated events. In general, one would expect the magnitudes of these differences to be small: although annotators clearly disagree on the precise number of events (see Table 1), the text will usually constrain them not to give substantially different counts. This is borne out in Table 5, which shows the ratio form of $\alpha$. As expected, the aggregate agreement is higher here for both datasets, though we stress that it is still far from what one would want from expert annotators ($\alpha$) and from the data ($\alpha^+$).

The high agreement observed on $\alpha^+$ among annotators for select MUC-4 template types suggests that some types of event (e.g. kidnappings) may just be more readily individuable than others — or perhaps merely that the documents annotated with these types happen to present especially clear-cut cases. We again emphasize that this does not impugn the broader point, supported by these results, that individuation is clearly challenging even for experts intimately familiar with the data.

Finally, concerning the observed drop in agreement for many template types when including the gold data, we acknowledge the possibility that this is largely due to differences in annotator training between our reannotation study and the original annotation. For instance, it's plausible that the original annotators informally agreed on certain annotation rules that were never made public, or resolved divergent annotations in dialogue among themselves. However, the possible existence of such consensus mechanisms also does not detract from the conclusions of our study, which explicitly interrogates how expert annotators *independently* approach this task, given the available documentation.

| BETTER | | | MUC-4 | | |
|---|---|---|---|---|---|
| **Type** | $\alpha$ | $\alpha^+$ | **Type** | $\alpha$ | $\alpha^+$ |
| **Corruption** | 0.23 | 0.27 | **Arson** | 0.48 | 0.35 |
| **Disaster** | 0.26 | 0.31 | **Attack** | 0.39 | 0.32 |
| **Epidemic** | 0.24 | 0.29 | **Bombing** | 0.95 | 0.68 |
| **Migration** | 0.09 | 0.07 | **Kidnapping** | 0.99 | 0.96 |
| **Protest** | 0.50 | 0.51 | **Robbery**[*] | 1.00 | 1.00 |
| **Terrorism** | 0.35 | 0.40 | | | |
| **Aggregate** | 0.33 | 0.38 | **Aggregate** | 0.68 | 0.66 |

Table 5: The ratio version of Krippendorff's $\alpha$, measuring IAA on the number of templates of each type for the reannotated MUC-4 and for BETTER documents.

| BETTER | | MUC-4 | | | |
|---|---|---|---|---|---|
| **Type** | ITERX | **Type** | ITERX | GTT | TEMPGEN |
| **Corruption**[*] | 0.00 | **Arson**[*] | -0.25 | -0.11 | 0.00 |
| **Disaster** | 0.24 | **Attack** | 0.08 | -0.01 | -0.14 |
| **Epidemic** | 0.64 | **Bombing** | 0.03 | 0.03 | 0.21 |
| **Migration** | -0.06 | **Kidnapping** | -0.20 | 0.14 | 0.00 |
| **Protest**[*] | 0.00 | **Robbery**[*] | 0.00 | 0.00 | 0.00 |
| **Terrorism** [*] | 0.00 | | | | |
| **Aggregate** | 0.25 | **Aggregate** | -0.05 | 0.16 | -0.02 |

Table 6: Nominal Krippendorff's $\alpha$ between gold template counts and counts as predicted by each model in §5 for all test set documents with at least one annotated template. Agreement is poor across the board, reinforcing our conclusion that template filling models do *not* effectively learn template count distributions. [*]Indicates that there are fewer than 10 documents in the test split containing annotated templates of this type and that the corresponding results should be read with caution.

# E   Model Details

The analysis in §5 is based on the GTT model released by Du et al. (2021b), which uses uncased BERT-base (Devlin et al., 2019) as its encoder.[13] The ITERX model is the one from Chen et al. (2023) that uses a T5-large encoder (Raffel et al., 2020), which showed the best results across the datasets presented in that paper. The TEMPGEN model is also from Chen et al. (2023), uses BART-large as the encoder (Lewis et al., 2020), and is lightly adapted from Huang et al. (2021) to support multi-template prediction.[14] Chen et al.'s source code is available here: `https://github.com/wanmok/iterx`.

To drive home the extent to which these models struggle to learn template count distributions (see §5), we report the level of agreement between their template count predictions on the MUC-4 and BETTER test sets and the template counts from the gold annotations in Table 6, computed on the subset of test set documents with at least one gold template of the corresponding type. Results for template types with minimal test set support (indicated by [*]) should be interpreted with caution, but agreement is generally very low and in many cases not much different from chance. With the caveat that the statistics in Table 1 and Table 6 are computed on different documents, it is noteworthy — and disheartening — that for BETTER, the level agreement between the gold annotations and ITERX (0.25; see Table 6) actually exceeds that between the gold annotations and the human experts (0.22; see Table 1).

---

[13]GTT model code and predictions are publicly available here: `https://github.com/xinyadu/gtt`.

[14]Model code and predictions for the original, unadapted TEMPGEN model are available here: `https://github.com/PlusLabNLP/TempGen`.

| Subevent Name | Phase | Subevent Description |
|---|---|---|
| Motivation | | The perpetrators formulate or express a motive or reason for carrying out the bombing. |
| Planning | | The perpetrators formulate plans for the bombing (e.g. selecting a target, working out the logistics).. |
| Preparations | Before | The perpetrators gather materials (e.g. the explosives) and information needed to carry out the bombing. |
| Transportation | | The perpetrators are transported, along with any necessary equipment, to the site of the bombing. |
| Placement | | The perpetrators place the explosive devices in the locations where they are later to be detonated. |
| Activation | During | The explosive devices that were placed detonate, possibly harming people or infrastructure. |
| Emergency Response | | Emergency personnel respond directly following the bombing. |
| Investigation | After | Authorities carry out an investigation into the bombing. |
| Apprehension | | The (suspected) perpetrators of the bombing are arrested or apprehended by authorities. |

Table 7: A possible subevent ontology for the MUC-4 `bombing` template.

## F  Subevent-Based Question Answering

Here, we sketch how our reframing of document-level event extraction as subevent-based QA (proposed in §7) could be carried out. As the aim of this paper is principally to highlight and explain the problem of event individuation, we stress that this is *a sketch*, and that a fully detailed solution is left for future work. We also think that subevent decompositions are valuable and interesting independent of our particular QA formulation; other approaches to subevent-based extraction are worth exploring.

To illustrate our proposal, we show how it could be implemented for the MUC-4 `bombing` event type. We imagine that the `bombing` event type could be broken down into the set of nine subevent types listed in Table 7. Many other subevent decompositions are possible; we present this one merely as an example. In §6, we note that the notion of a subevent decomposition we advocate for resembles *scripts* (Schank and Abelson, 1977) and *narrative schemas* (Chambers and Jurafsky, 2008, 2009), though we remain agnostic on certain questions of implementation where these proposals do not (e.g. we do not necessarily think subevents have to be associated with individual verbs, as events are in narrative schemas).

We demonstrate extraction against our subevent ontology on the document from Figure 1 using `gpt-3.5-turbo`, accessed via the OpenAI Playground.[15] For all queries, we use the following system prompt and the default Playground hyperparameters:

> You are very good at understanding and answering questions about events described in documents. I will ask you reading comprehension questions about a document and I want you to answer them directly and succinctly. Your answer should be based ONLY on what is explicitly said in the document. DO NOT answer based on what is merely implied.

We break the task down into a *subevent detection* step, in which we pose questions to determine which `bombing` subevent types are attested in the document, followed by *subevent identification* and *subevent argument extraction* steps, in which we pose questions to determine the number of instances of those subevent types and the arguments of each instance.[16] For subevent detection questions, the responses can generally be determined from the first word ("Yes" or "No"). But for subevent individuation and argument extraction, some parsing of the output would be required to pull out the responses; we leave this (or possible alternative question design) for future work — here just illustrating *that* ChatGPT can generally produce correct answers.

### F.1  Subevent Detection

We start by posing a series of questions, one per subevent type, aimed at determining which subevent types are attested in the document. In the spirit of *conditional* QA (§6), each question is conditioned on the complete dialogue history and more detailed questions — about the number of instances of a particular subevent type and the participants in those subevents — will later be conditioned on positive responses to these initial questions (more below). The article is presented as the initial context.

---

[15]https://platform.openai.com/playground

[16]Once again, this is by no means the only way to do things. For instance, subevent detection and identification could be rolled into one step.

Dialogue turns are indicated by **User** or **Assistant**. Subevents are queried in the order they are given in Table 7 and the subevent queried on a particular turn is indicated with purple // comments. By our own estimation, ChatGPT does this perfectly, correctly identifying that the document describes instances of both `placement` and `activation` subevents, but does *not* describe instances of any of the other types.

**User:** Article: "A bomb exploded today in a Lima restaurant, and a second device that had been placed in the same establishment was deactivated by the Peruvian National Police. There were no victims, and the explosion caused very little damage to the restaurant, which is located in the commercial area of the residential district of Miraflores. Guerrillas of the Tupac Amaru Revolutionary Movement (MRTA) have claimed credit for the terrorist act through pamphlets they left on the premises, according to the police."

Does this document explicitly describe any instances of a person or group expressing or formulating reasons for carrying out a bombing? // motivation

**Assistant:** No, the document does not explicitly describe any instances of a person or group expressing or formulating reasons for carrying out a bombing.

**User:** Does this document explicitly describe any instances of a person or group formulating plans to carry out a bombing, before actually doing so? // planning

**Assistant:** No, the document does not explicitly describe any instances of a person or group formulating plans to carry out a bombing before actually doing so.

**User:** Does this document explicitly describe a person or group gathering materials or information in preparation to carry out a bombing, before actually doing so? // preparations

**Assistant:** No, the document does not explicitly describe a person or group gathering materials or information in preparation to carry out a bombing before actually doing so.

**User:** Does this document explicitly describe any persons or groups being transported to some location to carry out a bombing? // transportation

**Assistant:** No, the document does not explicitly describe any persons or groups being transported to some location to carry out a bombing.

**User:** Does this document explicitly describe explosives being placed or planted in some target location? // placement

**Assistant:** Yes, the document explicitly describes a bomb exploding in a Lima restaurant and mentions that a second device was placed in the same establishment but was deactivated by the Peruvian National Police.

**User:** Does this document explicitly describe an explosive device going off or being triggered? // activation

**Assistant:** Yes, the document explicitly describes a bomb exploding in a Lima restaurant.

**User:** Does this document explicitly describe any emergency or support personnel responding to an explosion or the detonation of an explosive device? // emergency response

**Assistant:** No, the document does not explicitly describe any emergency or support personnel responding to an explosion or the detonation of an explosive device.

**User:** Does this document explicitly describe any persons or groups investigating an explosion or bombing? // `investigation`

**Assistant:** No, the document does not explicitly describe any persons or groups investigating an explosion or bombing.

**User:** Does this document explicitly describe any persons or groups being arrested or apprehended for having planted or activated an explosive device? // `apprehension`

**Assistant:** No, the document does not explicitly describe any persons or groups being arrested or apprehended for having planted or activated an explosive device.

### F.2  Subevent Individuation and Argument Extraction

Above, in the subevent detection step, ChatGPT (correctly) detected that two subevent types from the ontology, `placements` and `activations`, are attested in the document. We imagine that more detailed questions about these event types, including how many instances of them are described and who the participants are, would be conditioned on the relevant questions and (affirmative) answers from the subevent detection step.

For brevity, we illustrate this below for just one of the identified subevent types (`placement`). The full document text is again provided to the model as the initial context (omitted for brevity below), but here we also include the QA pair for the `placement` subevent from the subevent detection dialogue as additional input (all initial context is shown in gray). We assume we are interested in `PerpOrg`, `PerpInd`, `Weapon`, `Time`, and `Place` roles for `placement` events, though different role sets could of course be given. Following an initial query that determines the number of `placement` instances, we query for the arguments that fill these roles for each instance, one question per role, per identified instance. Additionally, we directly query for the irrealis status of the planting instances. Similar to above, queried roles are marked with // comments.

We find that ChatGPT does quite well in determining the number of `placement` events (2) and their irrealis status (both actual), although it gets more mixed results in identifying participants, failing to infer, for instance, that "MRTA" is the `PerpOrg` of the second `placement` event and refusing to identify the "guerrillas" as the `PerpInd` of both events.

**User:** Article: [*omitted; same as above*]

Does this document explicitly describe explosives being placed or planted in some target location?

**Assistant:** Yes, the document explicitly describes a bomb exploding in a Lima restaurant and mentions that a second device was placed in the same establishment but was deactivated by the Peruvian National Police.

**User:** How many distinct instances of explosives being placed or planted in some target location are explicitly described in this document?

**Assistant:** There are two distinct instances of explosives being placed or planted in the same target location explicitly described in this document.

**User:** Was a device actually placed in the first instance? // irrealis

**Assistant:** Yes, a device was actually placed in the first instance, as mentioned in the document.

**User:** What organization was responsible for the first instance? // `PerpOrg`

**Assistant:** The document states that the guerrillas of the Tupac Amaru Revolutionary Movement (MRTA) claimed credit for the first instance, which was the bomb explosion in the Lima restaurant.

**User:** What individuals were responsible for the first instance? // `PerpInd`

**Assistant:** The document does not explicitly mention the individuals responsible for the first instance. It only states that the guerrillas of the Tupac Amaru Revolutionary Movement (MRTA) claimed credit for the terrorist act through pamphlets they left on the premises.

**User:** What kind of device was planted in the first instance? // `Weapon`

**Assistant:** The document does not explicitly mention the kind of device that was planted in the first instance.

**User:** Where was the device planted in the first instance? // `Place`

**Assistant:** The document explicitly states that the device was planted in the same establishment, which is the Lima restaurant where the bomb exploded.

**User:** When was the device planted in the first instance? // `Time`

**Assistant:** The document does not explicitly mention when the device was planted in the first instance.

**User:** Was a device actually placed in the second instance? // irrealis

**Assistant:** Yes, a device was actually placed in the second instance, as mentioned in the document. However, it was deactivated by the Peruvian National Police.

**User:** What organization was responsible for the second instance? // `PerpOrg`

**Assistant:** The document does not explicitly mention the organization responsible for the second instance. It only states that the second device, which was placed in the same establishment, was deactivated by the Peruvian National Police.

**User:** What individuals were responsible for the second instance? // `PerpInd`

**Assistant:** The document does not explicitly mention the individuals responsible for the second instance.

**User:** What kind of device was planted in the second instance? // `Weapon`

**Assistant:** The document does not explicitly mention the kind of device that was planted in the second instance.

**User:** Where was the device planted in the second instance? // `Place`

**Assistant:** The document states that the second device was placed in the same establishment as the first instance. Therefore, the device was planted in the Lima restaurant where the bomb exploded.

**User:** When was the device planted in the second instance? // `Time`

**Assistant:** The document does not explicitly mention when the device was planted in the second instance.