# OpenReview forum: "On Event Individuation for Document-Level Information Extraction"
_EMNLP/2023/Conference — EMNLP 2023 Findings_

### Official Review · Reviewer_vPzg · 2023-07-31

**Soundness:** 4

**Excitement:**

4: Strong: This paper deepens the understanding of some phenomenon or lowers the barriers to an existing research direction.

**Paper Topic And Main Contributions:**

The position paper calls into question the usefulness of evaluating information extraction using template-filling metrics. The authors specifically focus on event individuation, the decision of whether one description may contain two individual events. Using both an annotation study and altered evaluation metrics the authors show that neither humans nor models are good at individuation events. Finally, the authors make some suggestions for alternative evaluation options.

**Questions For The Authors:**

Would you agree that the drop in agreement after the inclusion of the gold data could also indicate a difference in annotator training, perhaps something that was not captured properly in the annotation guidelines? This possibility is mentioned in reference to the models but should also be included in Section 4. The same goes for the availability of the BETTER annotation guidelines mentioned in the appendix.

What about cross-template individuation (e.g. attack and bombing), does this also have an effect on evaluation?

Collapsing the templates did not alter the relative performance of the three considered models (in terms of ranking), does that mean collapsed scores are not a helpful metric?

**Reasons To Accept:**

The paper calls into question popular evaluation techniques and datasets meaning it can potentially have a large impact on the field of information extraction.

The paper is very well written and clearly describes the problem through the use of an example, the line of argument is easy to follow.

Analsysis of prior work seems to have been extensive with detailed references to annotation guidelines being included in the appendix.

Three well-performing models are evaluated wrt their individuation capabilities, firmly establishing that SotA models are not capable of event individuation.

**Reasons To Reject:**

Scope and impact are limited to template filling IE

**Reproducibility:**

5: Could easily reproduce the results.

**Reviewer Confidence:**

3: Pretty sure, but there's a chance I missed something. Although I have a good feel for this area in general, I did not carefully check the paper's details, e.g., the math, experimental design, or novelty.

**Typos Grammar Style And Presentation Improvements:**

Table 2: consider typesetting the '|' symbols in math mode if they aren't already, I found them to slightly confusing at first.

---

> ### Author Rebuttal · Authors · 2023-08-25
>
> We thank Reviewer 3 for this thoughtful review and for the great questions!
>
> **Questions:**
>
> A. This is a great point. In short, yes — it's certainly possible that there was relevant context (private discussions on particular annotation questions, agreed-upon conventions) that the original annotators of these datasets had, that did not make it into writing, and that are therefore simply unavailable to us. We will explicitly note this possibility in our revisions.
>
> B. With ontologies like MUC and BETTER that have only a handful of event types that are fairly distinct, our experience suggests that individuation problems arising from cross-type confusions are generally much less of an issue than those arising from within-type confusions — hence our choice of focus in the paper. But this is likely a much more significant concern the more fine-grained the ontology is (e.g. FrameNet has roughly 1,200 frame types, some of which are very similar to each other) — another point we will note in our revisions.
>
> C. This is an astute question. See our response to Reviewer 2 on this point.
>
> **Reasons to Reject**
>
> We would dispute the implication that focusing on a particular NLP task (template filling) constitutes a reason for rejection; many effective papers do precisely this. Moreover, even if our analysis is focused on template filling, we think the core considerations about event individuation apply in any setting that (even implicitly) requires event coreference, which does not strike us as an especially niche problem.

---

### Official Review · Reviewer_HbG3 · 2023-08-04

**Soundness:** 3

**Excitement:**

4: Strong: This paper deepens the understanding of some phenomenon or lowers the barriers to an existing research direction.

**Paper Topic And Main Contributions:**

This position paper calls into question the usage of template filling as a benchmark for document-level information extraction. The authors argue that the problem with template filling is the lack of agreement on event individuation, and demonstrate the problem from 3 aspects. First, with an example that contains two events in the ground truth annotation but these events happen at the same time, location, and agents so the distinction of the two events is a bit arbitrary, the authors show the standard evaluation metrics heavily depend on alignment (matching the predicted instance of an event to the instance in the ground truth). Second, the authors perform annotations themselves and show low inter-judge agreement as well as low agreement with the ground truth annotation. Finally, the authors show that state-of-the-art IE models have a degenerate behavior: they collate all entities relevant to a particular event type into a single template. The implication of this paper is that the current benchmark may be misleading for document IE.

**Reasons To Accept:**

This paper is well-written. It proposes an important issue: the current dataset and benchmark can be misleading. I'm wondering if the 3 IE models in Section 4 demonstrate the degenerate behavior of collating all entities relevant to a particular event type into a single template because they all overfit the existing benchmark. Also, if there exists another model, worse under the standard metrics but does not have the degenerate behavior, actually performs better if evaluated using a different approach, for example, using human evaluation or evaluating by metrics related to its real-world application. If this is true, then this paper is really important as the current research efforts may be working in the wrong direction.

**Reasons To Reject:**

The claim in this paper is important, but I cannot be 100% sure it is correct. It can be the case that some of the event individuation can be arbitrary and therefore the annotation is noisy, but the benchmark can still reflect the performance of the IE model. For example, as a future direction, the authors propose the possiblity of aggregate the template by type. However, in Table 2, the original Combine Score and the Collaposed Combine Score are highly correlated and give the same ranking for the 3 models. Despite showing the problem with annotations themselves, there is a lack of direct evidence that such noisy annotation fail to rank model's performance correctly.

**Reproducibility:**

4: Could mostly reproduce the results, but there may be some variation because of sample variance or minor variations in their interpretation of the protocol or method.

**Reviewer Confidence:**

2: Willing to defend my evaluation, but it is fairly likely that I missed some details, didn't understand some central points, or can't be sure about the novelty of the work.

---

> ### Author Rebuttal · Authors · 2023-08-25
>
> We thank Reviewer 2 for this careful and very thoughtful review! We're glad our main thesis seems important, and we hope to resolve doubts about its correctness here.
>
> *It can be the case that some of the event individuation can be arbitrary and therefore the annotation is noisy, but the benchmark can still reflect the performance of the IE model....Despite showing the problem with annotations themselves, there is a lack of direct evidence that such noisy annotation fail to rank model's performance correctly.*
>
> This is a shrewd observation. However, our aim isn't necessarily to show that the ranking of the models actually induced by the metrics is definitely wrong. Rather, our goals are to show that:
>
> 1. Each model, considered independently, seems to struggle to do individuation *at all* (as evidenced by results in Table 2).
> 2. Even experts disagree in many cases on what the "correct" individuation is.
> 3. The standard benchmarks (datasets + metrics) *will* yield quite divergent results for similar predictions that differ mainly in their individuation decisions (as evidenced by the discussion in section 3).
>
> We gather Reviewer 2 agrees with these claims, but we don't think they are undermined by the above critique.
>
> Part of the critique seems to say: *if (3) is true, we should see clearer evidence for it in the empirical results — e.g. in the form of a reranking of the models in Table 2 under the collapsed scores*. Why don't we observe this? We argue that it's in fact largely because of (1): if all our best models still tend overwhelmingly to predict one template per type (when they ought to predict more), not much *should* change in the results — likely including their relative ranking — when moving to collapsed scores. This is exactly the point we make in section 5.
>
> But when *would* we expect (3) to be evidenced more clearly by the empirical results? Presumably, it would be when models actually *do* demonstrate an ability to effectively predict multiple templates of the same type. Here, we might expect to see more variability in models’ individuation decisions. Unfortunately, as we show, even SOTA models fail to demonstrate this ability. But in L253-6, we try to gesture at what would be at stake if we *did* have better models by showing the percentage of a perfect score that hinges on getting individuation "right" (per the benchmark). These percentages (12.7% for MUC, 24.2% for BETTER) are substantial and they suggest one could very well see a reranking of models under collapsed scores — if only the models were better than they are.
>
> In sum, it could be that the metrics happen to rank *these* models "correctly" (whatever this means, independent of metrics), but this isn't evidence against (1)-(3), nor against the broader claim that individuation poses a significant challenge — even for humans and surely, too, for better models down the line. At most, it could be evidence that the standard metrics are adequate for model comparison purposes so long as those models do minimal individuation.

---

### Official Review · Reviewer_yycY · 2023-08-04

**Soundness:** 4

**Excitement:**

4: Strong: This paper deepens the understanding of some phenomenon or lowers the barriers to an existing research direction.

**Paper Topic And Main Contributions:**

This position paper critically examines the benchmark task of template filling for document-level information extraction. The main problem it addresses is that template filling relies on arbitrary decisions about how to individuate distinct events, even though experts often disagree about individuation.

The key contributions are:

- Demonstrates how standard template filling metrics penalize different individuation choices even when all entities and roles are correctly extracted. This harms comparability between models.

- Presents an annotation study showing that even experts struggle to agree on event individuation decisions. Suggests common datasets embody questionable individuation judgments.

- Analyzes state-of-the-art models, finding they fail to learn the individuation rules latent in datasets and heavily under-generate templates.

- Proposes recasting template filling as question answering without requiring explicit individuation decisions. Argues key event properties like time and location should be queried directly.

- Provides critical analysis and evidence that individuation decisions are often subjective, unreliable, and detrimental as part of the template-filling benchmark.

**Reasons To Accept:**

- Provides a compelling critique on an important IE benchmark task, raising awareness about a neglected issue that could stimulate useful discussion.

- Backs its central claims with rigorous empirical evidence from multiple angles - annotation studies, model analysis, and example case studies. The methodology is thorough.

- Focuses attention on an aspect of IE that has major room for improvement - event individuation. The analysis reveals the limitations of current benchmarks and systems.

- Proposes an interesting alternative QA-based formulation that could better handle individuation while avoiding arbitrary decisions. This explores a novel direction.

- Evaluates established datasets like MUC-4 and BETTER. Demonstrates shortcomings exist across datasets and state-of-the-art models.

- Well-written and structured. Clearly articulate the position and line of argumentation.

**Reasons To Reject:**

I do not see any major weaknesses or risks that would warrant rejecting this paper based on the conference policies. The paper provides a cogent position supported by thorough analysis. However, some aspects that could be improved:

- The alternative QA formulation is only briefly sketched. More details are needed to demonstrate its feasibility as a full solution.

- The paper is very short at only 4 pages, with a 12-page appendix containing important details on metrics, proposed approach, etc. This suggests key content is relegated to the appendix rather than appearing in the main paper body. Significant details around the QA proposal and metric definitions are covered in the appendix rather than the main text. This makes the core ideas and innovations harder to understand on their own. The rationale for the appendix length compared to the main paper is unclear. It seems excessive, suggesting the contribution could be better communicated as a long paper.

**Reproducibility:**

N/A: Doesn't apply, since the paper does not include empirical results.

**Reviewer Confidence:**

3: Pretty sure, but there's a chance I missed something. Although I have a good feel for this area in general, I did not carefully check the paper's details, e.g., the math, experimental design, or novelty.

---

> ### Author Rebuttal · Authors · 2023-08-25
>
> We thank Reviewer 1 for this encouraging review! We're pleased both the paper's core thesis and supporting arguments seem compelling.
>
> As this is a position paper, and as the primary aim is to convince the reader of the challenges arising from event individuation, we wanted to devote the majority of allotted space to arguments in support of this claim. However, we certainly agree that the proposed QA formulation merits further elaboration. Should the paper be accepted, we intend to use the additional page for this purpose. In particular, we think a framework that outlines how these question sets could be constructed (beyond the one example in Appendix F) is warranted.
>
> We also understand and appreciate the comment on paper vs. appendix length. We felt that the core arguments for the thesis could be effectively communicated in a short paper, and we stand by this decision. While the appendices do run long, we do *not* think they are necessary to fully grasp the content in the main text, and are rather intended to:
>
> 1. Provide additional helpful, though inessential, context that could largely be found by reading the primary citations in the main text (Appendices A and B)
> 2. Preempt concerns that we might have cherry-picked our example in Figure 1 by providing additional examples of cases that pose challenging individuation decisions (Appendix C).
> 3. Promote good science by sharing further details of our methods and models (Appendices D and E) that the average reader likely won't care to know but that are important to include all the same.
>
> We feel the great majority of this content really does belong in an appendix, rather than the main text. A plausible exception to this is Appendix F on the QA formulation, which we have addressed above, and which we would like to discuss at greater length with the additional page. That said, we would be happy to trim any parts of the appendices that seem superfluous.

---

### Meta-Review · Area_Chair_7wcf · 2023-09-20

**Recommendation:** 3

**Metareview:**

The paper proposes the use of an alternative evaluation metric for document-level information extraction, instead of relying on the existing method of template filling, which depends on arbitrary decisions about how to distinguish distinct events. Such distinction is challenging for humans and can also pose difficulties for models, as it is often motivated by subjective and unreliable factors, as indicated by an annotation studies. The alternative solution proposed is cast the problem as a question-answering task.

The paper is well-written, offering insightful observations about a neglected issue that can stimulate discussions. Furthermore, the methodology is thorough, and the approach of framing it as a question-answering problem represents a novel direction. The evaluation is conducted on the MUC-4 and BETTER datasets.

---

### Decision · Program_Chairs · 2023-10-07

**Decision:**

Accept-Findings

**Comment:**

The paper proposes the use of an alternative evaluation metric for document-level information extraction, instead of relying on the existing method of template filling, which depends on arbitrary decisions about how to distinguish distinct events. Such distinction is challenging for humans and can also pose difficulties for models, as it is often motivated by subjective and unreliable factors, as indicated by an annotation studies. The alternative solution proposed is cast the problem as a question-answering task.

The paper is well-written, offering insightful observations about a neglected issue that can stimulate discussions. Furthermore, the methodology is thorough, and the approach of framing it as a question-answering problem represents a novel direction. The evaluation is conducted on the MUC-4 and BETTER datasets.